# Emerging SARS-CoV-2 Variants and Impact in Global Vaccination Programs against SARS-CoV-2/COVID-19

**DOI:** 10.3390/vaccines9030243

**Published:** 2021-03-11

**Authors:** Carmen Elena Gómez, Beatriz Perdiguero, Mariano Esteban

**Affiliations:** Centro Nacional de Biotecnología, Department of Molecular and Cellular Biology, Consejo Superior de Investigaciones Científicas (CNB-CSIC), Campus de Cantoblanco, 28049 Madrid, Spain; perdigue@cnb.csic.es

**Keywords:** SARS-CoV-2, COVID-19, variant, spike protein, mutation, lineage, vaccine efficacy, neutralizing antibodies

## Abstract

The emergence of severe acute respiratory syndrome coronavirus-2 (SARS-CoV-2) variants in different continents is causing a major concern in human global health. These variants have in common a higher transmissibility, becoming dominant within populations in a short time, and an accumulation of a high number of mutations in the spike (S) protein, especially within the amino terminal domain (NTD) and the receptor binding domain (RBD). These mutations have direct implications on virus infection rates through higher affinity of S RBD for the cellular angiotensin-converting enzyme-2 (ACE-2) receptor. There are also signs of enhanced virulence, re-infection frequency, and increased resistance to the action of monoclonal and polyclonal antibodies from convalescence sera and in vaccinated individuals in regions where the variants spread dominantly. In this review, we describe the different SARS-CoV-2 variants that have thus far been identified in various parts of the world with mutational changes and biological properties as well as their impact in medical countermeasures and human health.

## 1. Introduction

Severe acute respiratory syndrome coronavirus-2 (SARS-CoV-2) is the causal agent of the worldwide coronavirus disease 2019 (COVID-19) pandemic, which is causing major health as well as social and economic burden with unprecedented consequences. Records show 116 million infection cases worldwide with a global death of 2.6 million people early March 2021. At the beginning of the SARS-CoV-2 pandemic, there were only modest levels of genetic evolution mainly because of two factors: (i) the global absence of immunity against this new pathogen; and (ii) the low mutation rates of the coronaviruses which encode an enzyme with proofreading function that increases the fidelity of the replication process [1]. In early March 2020, a new variant was detected with a single D614G mutation in the spike (S) glycoprotein of SARS-CoV-2 that spread to global dominance over the next month due to increased transmissibility and virus replication [2,3]. Since December 2020, novel SARS-CoV-2 variants that accumulate a high number of mutations, mainly in the S protein, have been detected in some geographical regions. These variants have been considered by the World Health Organization (WHO) as variants of concern (VOC) because of their potential risk to human health. The changes observed in the viral mutation rate during the course of the pandemic indicate a tendency towards a rapid antigenic variation and, hence, it is important to strengthen surveillance systems to control the emergence and the dissemination of new variants, looking over their impact on disease transmission and severity and on the efficacy of vaccines and treatments used globally.

The COVID-19 pandemic has required rapid action and development of vaccines in an unprecedented timeframe. According to WHO (https://www.who.int/publications/m/item/draft-landscape-of-covid-19-candidate-vaccines (accessed on 9 March 2021)), 76 vaccine candidates based on several different platforms are being currently evaluated in human clinical trials, while 182 candidates are under investigation in preclinical models. The four SARS-CoV-2 vaccines licensed at present by the regulatory agencies are based on nucleic acid or non-replicating viral vectored platforms.

The two vaccines based on messenger ribonucleic acid (mRNA) have been developed by Moderna (mRNA-1273) and Pfizer/BioNTech (BNT162b2) pharmaceutical companies and contain the genetic information for the synthesis of the stabilized pre-fusion form of the SARS-CoV-2 spike (S) protein encapsulated in a lipid nanoparticle (LNP) vector that enhances uptake by host immune cells. These vaccines used host cell transcription and translation machinery to produce the viral S protein, that is afterwards processed and recognized by specific B and T cells eliciting both humoral and cellular adaptive immune responses able to confer protection against COVID-19 illness, including severe disease. The reported efficacy of a two-dose regimen of the mRNA-1273 or the BNT162b2 vaccines is 94.1% [4] or 95% [5], respectively.

The two other licensed vaccines have been developed by Oxford University/AstraZeneca (AZD1222) and Janssen (Ad26.COV2.S) pharmaceutical companies and are based on two different modified non-replicating adenoviruses. The AstraZeneca candidate is a monovalent vaccine composed of a single recombinant, replication-deficient chimpanzee adenovirus (ChAdOx1) vector encoding the S glycoprotein of SARS-CoV-2. The S protein is expressed in the trimeric pre-fusion conformation. Following administration, the S glycoprotein of SARS-CoV-2 is expressed locally, stimulating neutralizing antibody and cellular immune responses, which may contribute to protection against COVID-19. The AZD1222 vaccine has an efficacy of 63.09% against symptomatic SARS-CoV-2 infection. Vaccine efficacy was 62.6% in participants receiving two recommended doses with any dose interval (ranging from 3 to 23 weeks) [6]. The Janssen vaccine is based on the adenovirus serotype 26 (Ad26) which expresses the stabilized pre-fusion SARS-CoV-2 S protein. As opposed to the ubiquitous Ad5 serotype, very few people have been exposed to the rare Ad26 serotype; therefore, pre-existing immunity against the vector reducing this candidate’s immunogenicity is not expected to be a major concern. A phase 3 randomized and placebo-controlled trial of the single-dose Ad26.COV2.S in approximately 40,000 participants is currently ongoing. The primary analysis of 39,321 participants using a data cut-off date of 22 January 2021 demonstrated a vaccine efficacy of 66.9%.

The fifth vaccine waiting for approval has been developed by Novavax Company (NVX-CoV2373). It is a protein subunit vaccine constructed from the full-length, stabilized, pre-fusion SARS-CoV-2 S glycoprotein, produced in the established Sf9 insect cell expression system and adjuvanted by saponin-based Matrix M1 [7]. In January, Novavax announced that, in the British trial, the vaccine had an efficacy rate of 89%. Since all the vaccines that have been administered worldwide are focused on the spike protein, which accumulates high rate of mutations during viral evolution, as evidenced in the genome sequences from the new emerging SARS-CoV-2 variants, it is imperative to evaluate the impact of those mutations on the actual efficacy of COVID-19 vaccines.

## 2. Emerging SARS-CoV-2 Variants That Raise Global Concern

Natural selection usually determines the fate of a newly arising mutation. However, other potential mechanisms such as chance events, host shifts, persistent infection in immunocompromised host, or mutations affecting the proofreading function could also drive viral evolution. Those mutations that confer a competitive advantage with respect to viral replication, transmission, or escape from immunity will increase in frequency, becoming the dominant variant. Currently, an exhaustive surveillance monitoring has being performed on three SARS-CoV-2 variants: B.1.1.7 (VOC 202012/01 or 20B/501Y.V1), B.1.351 (20H/501Y.V2), and P.1 (B.1.1.28.1).

### 2.1. B.1.1.7 (VOC 202012/01 or 20B/501Y.V1) Variant

B.1.1.7 variant, also known as VOC 202012/01 or 20B/501Y.V1, was unveiled on 14 December 2020 by the United Kingdom (UK) authorities who declared an increase in the incidence of SARS-CoV-2 infections in eastern and south-eastern England and the London metropolitan area associated with the emergence of a new SARS-CoV-2 variant. The first clinical sample in which it could be identified dates back to September 2020. Since then, it has largely replaced the circulating viruses, becoming the predominant variant in the UK. This variant is characterized by greater transmission, which has contributed to an increase in incidence, hospitalizations, and pressure on the health system since the second half of December 2020. Epidemiological studies and mathematical modeling suggest that it spreads 56% faster than other lineages and results in higher nasopharyngeal viral loads than the wild-type strain [8]. By January 2021, the UK reported the highest daily mortality from COVID-19 since the onset of the pandemic [9]. Retrospective observational studies estimate a 35% (12–64%) increased risk of death associated with B.1.1.7 variant, indicating that it is not only more transmissible than pre-existing linages but can also cause a more serious disease [10]. However, there was no evidence of more severe disease in children and young people [11].

Compared to ancestral viruses containing the D614G mutation, the B.1.1.7 variant has accumulated 23 mutations, and it is not phylogenetically related to the viruses circulating in the UK when it was detected. Of these mutations, 14 are non-synonymous: [[T1001I, A1708D, and I2230T] in open reading frame (ORF)1ab; [N501Y, A570D, P681H, T716I, S982A, and D1118H] in the spike (S) protein; [Q27stop, R52I, and Y73C] in ORF8; and [D3L and S235F] in the nucleocapsid (N) protein]; 6 are synonymous: [[C913T, C5986T, C14676T, C15279T, and T16176C] in ORF1ab; and [T26801C] in M (membrane) gene]; and 3 are deletions: [[SGF 3675-3677del] in ORF1ab; and [H69-V70del and Y144del] in S protein]. It is unclear how this variant achieves prominence, although the unusual genetic divergence of the B.1.1.7 lineage may have resulted, at least in part, from the evolution of the virus in an individual with chronic infection [12]. Excluding synonymous mutations, 47% of reported changes in B.1.1.7 variant occur in the gene encoding for the spike protein that contains the receptor binding domain (RBD). SARS-CoV-2 S protein is the main target of neutralizing antibodies and, hence, it has been used as vaccine antigen in most of the SARS-CoV-2 candidates under development and in the licensed vaccines that are being globally administered. The high frequency of mutations in this protein has caused global concern because these mutations, either individually or as a whole, can induce structural changes that might: (i) alter the interaction with the human angiotensin-converting enzyme-2 (hACE2) receptor, modifying the infection rate or even the interaction of the virus with the ACE2 receptor from other species supporting host shifts; (ii) modify the efficacy of both neutralizing antibodies and specific T cells elicited either during natural infection or through vaccination; or (iii) alter the sensitivity to neutralization by monoclonal antibodies or sera from convalescent patients, compromising the efficacy of treatments. The three mutations of B.1.1.7 with the greatest potential to affect the biological behavior of the virus are: H69-V70del, N501Y, and P681H. H69/V70 deletion is one of the recurrent mutations observed in the amino terminal domain (NTD) of S protein and emerged independently in at least six lineages of the SARS-CoV-2 virus prevalent in Europe. It is present in over 6000 sequences worldwide and often co-occurs with the RBD amino acid replacements N501Y, N439K, and Y453F [13,14]. H69/V70 deletion itself favored a two-fold increase in S protein-mediated infectivity in vitro using pseudotyped lentivirus, indicating that this effect on virus fitness seems to be independent of the RBD changes [13]. Protein structure modeling shows that H69/V70 deletion could be a “permissive” mutation that modifies the immunodominant epitopes located at variable loops within NTD, conferring resistance to neutralization by sera from both convalescent patients and vaccinated individuals [14]. The N501Y mutation is of major concern because it involves one of the six key amino acid residues determining a tight interaction of the SARS-CoV-2 RBD with ACE2 receptor. Modeling analysis showed that the N501Y mutation would allow a potential aromatic ring-–ring interaction and an additional hydrogen bond between RBD and ACE2 [15] and, hence, an increase in the binding affinity of SARS-CoV-2 S protein for hACE2 receptor [16]. A retrospective study has recently reported a three-fold higher inferred viral loads in a group of UK individuals infected with the viral variant carrying the N501Y mutation, evidencing the high efficiency of infection and transmission associated with B.1.1.7 variant [17]. The N501Y mutation has also been associated with increased infectivity and virulence in mouse and ferret models [18]. The function of the P681H mutation is not yet clear, but it is located immediately adjacent to the amino acids 682–685, the furin cleavage site (FCS) identified at the S1/S2 in the spike protein. SARS-CoV-2 FCS is not found in closely related coronavirus and has been shown to promote the entry of the virus into respiratory epithelial cells [19]. Similarly, it has been shown that the insertion generated by FCS into SARS-CoV-2 S protein enhances transmembrane serine protease (TMPRSS)-induced cleavage ability and viral infectivity [20]. Both N501Y and P681H mutations have been reported independently, but thus far they have not appeared in combination.

The mutation Q27stop in ORF8 observed in the B.1.1.7 variant truncates the ORF8 protein or makes it inactive, allowing the accumulation of additional mutations in other regions. At the beginning of the pandemic, multiple viruses with deletions that led to the loss of ORF8 expression were isolated worldwide, including a large group in Singapore. These deletions were associated with a milder clinical infection and lower post-infection inflammation; however, this group disappeared at the end of March after Singapore successfully implemented control measures. In subsequent reports, it was found that the deletion of ORF8 has only a very modest effect on virus replication in human primary respiratory cells compared to viruses without the deletion [21].

The rapid transmission of B.1.1.7 variant in and out of the UK suggests that this variant could become the dominant lineage responsible for the upcoming infections in Europe. As of 16 February 2021, 71.413 sequences of B.1.1.7 lineage have been detected in 64 countries, while 82 have reported cases related with this variant [22]. This emphasizes the importance of a global approach to surveillance, tracking, and vaccine deployment.

How does B.1.1.7 dissemination impact the efficacy of vaccines and treatments that are being administered globally? To date, low or no significant impact of the vaccines efficacy against B.1.1.7 variant has been reported. Xiu et al. show small effects of some of the mutations present in the B.1.1.7 variant on neutralization by sera from vaccinated individuals elicited after two BNT162b2 doses (Pfizer). Nevertheless, the modified virus used in the study lacked the full set of spike mutations described in B.1.1.7 variant [23]. Similarly, Collier et al. did not observe a significant reduction in the ability of sera from vaccinated individuals (Pfizer) to inhibit parental or mutant pseudoviruses including only three (H69/V70del, N501Y, and A570D) S mutations; however, a reduction in the neutralization titers was evident using pseudoviruses with the complete set of mutations described for B.1.1.7 variant [24]. Slightly reduced but overall largely preserved neutralizing titers against the B.1.1.7 lineage pseudovirus were also detected in sera from people receiving the Pfizer vaccine [25]. Correspondingly, modest reductions in the neutralizing activity of both plasma from convalescent patients (2.7–3.8-fold) and sera from individuals that received Moderna or Pfizer vaccines (1.8–2-fold) were described by Wang et al. [26]. This group also evaluated a panel of 30 monoclonal antibodies (mAbs) directed against the NTD and the RBD domains of S protein and observed that the B.1.1.7 variant is refractory to neutralization by most of the mAbs that recognized NTD and is relatively resistant to several RBD-directed mAbs, including two that have been approved for emergency treatment [26]. Finally, Wu et al. did not detect a significant impact in the neutralizing capacity of sera from human subjects or non-human primates (NHPs) who received the mRNA-1273 (Moderna) vaccine against the B.1.1.7 variant [27]. Overall, although these in vitro studies have their limitations either by methodology, by sample size, or by only considering the humoral arm of the immune response, what they indicate is that the efficacy of the vaccines that have being administered is similar or moderately lower against the B.1.1.7 variant.

### 2.2. B.1.351 (20H/501Y.V2) Variant

The second variant that has raised global concern is the B.1.351 variant, also known as 20H/501Y.V2, identified on 18 December 2020 by the authorities from Republic of South Africa. The first clinical sample in which this variant could be detected dates back to 8 October 2020, and one month later, it replaced the circulating viruses in the region, becoming the dominant variant. This behavior suggests higher transmission rates, although no evidence of greater virulence or disease severity has been reported to date. As of 16 February 2021, 1383 sequences of B.1.351 lineage have been detected in 35 countries, while 40 have reported cases related with this variant [22].

Compared to the Wuhan reference strain, the B.1.351 variant has 12 non-synonymous mutations and one deletion. Approximately 77% of these mutations are located in the spike protein [L18F, D80A, D215G, LAL 242–244 del, R246I, K417N, E484K, N501Y, D614G, and A701V] while the remaining ones are located in ORF1a [K1655N], envelope (E) [P71L], and N [T205I] viral proteins. Similar to B.1.1.7 variant, it could have emerged through intra-host evolution in one or more individuals with prolonged viral replication. However, the high number of mutations accumulated within two of the most immunodominant regions of S protein, such as the NTD and the RBD domains, suggests that it could also be originated as an escape variant to neutralization [28]. Therefore, the concern about the effect that the B.1.351 mutations might have on vaccine efficacy and treatments is even greater than with the B.1.1.7 variant.

B.1.351 and B.1.1.7 variants share the N501Y mutation, located in the RBD domain of the spike protein. As described above, this mutation confers an increased binding affinity of S RBD for the ACE2 receptor, raising the viral transmission rate. However, in addition to the N501Y mutation, this variant accumulates two additional mutations in the same RBD domain (K417N and E484K) that could play a pivotal role in both the interaction with the receptor and the immune evasion. The E484K mutation is present in <0.02% of sequences outside South Africa. There is evidence that this mutation can modestly improve the binding affinity of the virus to the receptor. Similarly, although K417 residue is a unique SARS-CoV-2 S RBD residue that interacts with ACE2 contributing to an enhanced affinity of the virus for the receptor, a mutational scanning study suggests that the amino acid change of K by N minimally affects this binding [16]. Moreover, a molecular dynamics simulation study reveals that both E484K and N501Y mutations increase affinity of S RBD for hACE2 and E484K in particular switches the charge on the flexible loop region of S RBD, which leads to the formation of novel favorable contacts. Moreover, the combination of E484K, K417N, and N501Y mutations results in the highest degree of conformational alterations of S RBD when bound to hACE2, compared to either E484K or N501Y alone, allowing the virus a more effective escape to neutralization [29]. It should be noted that the rest of the mutations that this variant holds in the spike protein are located in the NTD region (L18F, D80A, D215G, LAL 242-244 del, and R246I), especially within or near flexible variable loops that accept sequence changes without modifying the structure of the functional domains of S protein. NTD is also a preferential target of antibodies isolated from convalescent patients or vaccinated individuals. Preliminary studies highlight that a combination of RBD and NTD mutations in the B.1.351 spike protein significantly affects the neutralization of this variant by both mAbs targeting these regions and immune sera derived from convalescent or vaccinated patients [26,27,30,31,32,33]. Studies evaluating panels of potent mAbs that have been identified against the spike protein indicate that B.1.351 variant is refractory to be neutralized by antibodies that recognize S NTD while the S RBD mutations reduce or abolish the neutralization capacity of nearly 80% of the mAbs that recognize this region [26,30,31]. In line with these findings, a significant reduction in the neutralization capacity of polyclonal antibodies derived from convalescent patients against B.1.351 variant has been reported, being the sera from hospitalized patients with a more severe disease those who displayed higher neutralization efficacy [26,31,32,33]. These data point to a possible decrease in the efficacy of the treatments based on monoclonal or polyclonal antibodies as well as an increase in the rate of reinfection in regions where this variant spreads dominantly.

In terms of vaccine efficacy against B.1.351 variant, the results are more worrying than those shown for B.1.1.7. Wang et al. show that B.1.351 variant is notably more resistant to neutralization by polyclonal antibodies derived from individuals vaccinated with Pfizer (6.5-fold) or Moderna (8.6-fold) vaccines [26]. Furthermore, Wu et al. reported a significant decrease in the neutralization of B.1.351 variant by sera from either humans or NHPs who were vaccinated with mRNA-1273 (Moderna); however, the authors highlight that, despite the observed decreases, the geometric mean of the neutralization titers in sera from human vaccinated individuals against B.1.351 variant remained at ~1/300, a value that they still consider acceptable for protection [27]. Finally, a new study assaying the antibody responses and memory B cells in volunteers who received Moderna (mRNA-1273) or Pfizer (BNT162b2) vaccines reported a reduced activity against the SARS-CoV-2 variants that contain the E484K or the N501Y mutations or the K417N:E484K:N501Y combination [31]. Overall, these data reflect a more pronounced decrease in the efficacy of antibody-based vaccines and therapies against this variant.

The most worrying statistics are derived from the interim efficacy results from two randomized placebo-controlled clinical trials reported in press release by Novavax and Janssen companies in South Africa, the region where B.1.351 variant prevails. Biotechnology company Novavax reports that its COVID-19 vaccine NVX-CoV2373, which includes the S protein from the SARS-CoV-2 Wuhan reference strain, has shown 85.6% efficacy against the B.1.1.7 variant (95.6% against the original strain) in the phase 3 clinical trial involving 15,000 participants between 18 and 84 years old in the UK. However, in the phase 2b study developed in South Africa, involving more than 4400 individuals, the efficacy in the overall population was 49.4% against B.1.351 variant. This value increases up to 60% efficacy in the prevention of mild, moderate, and severe COVID-19 disease if the human immunodeficiency virus (HIV) positive group is eliminated from the overall count [34]. These interim data evidence a significant decrease in the efficacy of a vaccine influenced by the dominance of a viral variant such as B.1.351. Additionally, Janssen’s coronavirus vaccine has shown 72% efficacy in a single dose in the ENSEMBLE trial in the United States to prevent moderate to severe COVID-19 at 28 days after vaccination. However, these values were reduced to 66% in Latin America and 57% in South Africa. Despite the reduced efficacy, the rAd26 vaccine was 85% effective overall in preventing severe COVID-19, and protection was similar in all regions [35]. Finally, a clinical trial evaluating two-dose regimen of AZD1222 (AstraZeneca/Oxford vaccine) in South Africa did not show protection against mild to moderate COVID-19 due to B.1.351 variant [36]. Altogether, these results confirm that it is imperative to minimize the circulation of the virus, prevent infections, and reduce the opportunities for the SARS-CoV-2 to evolve, resulting in mutations that could reduce the efficacy of existing vaccines.

### 2.3. P.1 (B.1.1.28.1) Variant

The third variant of SARS-CoV-2 that raises concerns is P.1 variant, also known as B.1.1.28.1. It was detected by Japan’s National Institute of Infectious Diseases on 6 January 2021 and was isolated from four travelers who arrived in Tokyo from Amazonas, Brazil, on 2 January 2021 at airport control. P.1 variant was later identified in Brazil, where it has become the dominant circulating virus [37]. The rapid increase in the number of hospital admissions by COVID-19 in January 2021 (six-fold higher than the number reported in December) [38] is unexpected and worrying considering that this city reached 76% seroprevalence during the summer wave [39].

P.1 variant belongs to the B.1.1.28 lineage and contains 17 non-synonymous mutations: [L18F, T20N, P26S, D138Y, R190S, K417T, E484K, N501Y, D614G, H655Y, T1027I, and V1176F] in S protein, [S1188L, K1795Q, and E5665D] in ORF1ab, [E92K] in ORF8, and [P80K] in N protein; 1 deletion: [SGF 3675-3677del] in ORF1ab; and 4 synonymous mutations. P.1 is the SARS-CoV-2 variant that accumulates the highest number of mutations in the spike protein (12 mutations). The mutation N501Y is present in the three variants, while L18F, K417T, E484K, and D614G mutations are shared with the B1.351 variant. As described above, this set of spike mutations has important implications for transmission, reinfection rates, and evasion of antibody-mediated immunity. In fact, one clinical case of reinfection has been reported in this region [40]. As of 16 February 2021, 150 sequences of P.1 lineage have been detected in 14 countries, while 18 have reported cases related with this variant [22].

One of the most worrying mutations in terms of immune evasion is the E484K, which is shared by the P.1 and the B.1.351 variants. Recently, the effect of this mutation has been evaluated in the neutralization ability of sera from convalescent or vaccinated patients considering their SARS-CoV-2 spike immunoglobulin G (IgG) antibody titer. The efficacy of serum neutralization against the virus carrying the E484K mutation was reduced in both vaccination samples and convalescent sera. However, sera with high anti-S IgG titers were still able to neutralize the virus with the mutation, indicating that it is important to induce the highest possible levels of specific antibodies through vaccination to improve protection against emerging variants of SARS-CoV-2 [41]. Although this study does not use a virus containing the entire set of mutations that are present in combination with E484K in the spike protein of the different variants in order to provide a more realistic estimate of this effect, it makes clear that it is necessary to optimize the vaccination schedules to increase the possibility of counteracting the expansion of the new variants. Findings by Planas et al. using authentic clinical viral isolates to assess inherent viral fitness and potential impact of additional mutations outside of the spike on sensitivity to neutralizing antibodies also demonstrated that low global antibody levels or declining antibody responses are associated with a loss of cross-reactivity against novel emerging variants [42]. These findings indicate that it is necessary to follow rigorously the vaccination regimen established and approved by the regulatory authorities for the different licensed vaccines. In addition, it is important to study in the clinical settings how the introduction of combined or “prime/boost” heterologous vaccination protocols could optimize the strength of both humoral and cellular immune responses [43].

There is still a long way to go into the study of this and other variants circulating in Brazil; however, considering the high number of mutations that P.1 accumulates in the spike protein, it is probable that it behaves as resistant or even more resistant than B.1.351 variant to neutralization by monoclonal antibodies and vaccinee sera.

Scientists in Brazil reported on 14 January 2021 that the coronavirus vaccine of the Chinese pharmaceutical company Sinovac (CoronaVac) based on inactivated virus was 50.38% effective when tested in 12,508 volunteers, all of them health professionals in direct contact with the coronavirus [44]. It remains to be determined whether the efficacy of this vaccine (which is close to the approval threshold for emergence use) is maintained against the new P.1 variant that is expanding dramatically in the country.

A summary of the key features of the emergent SARS-CoV-2 variants is depicted in Table 1.

## 3. Other Variants of Interest

The United States is the country with the highest incidence rates of COVID-19, and different states have reported the prevalence of all the emergent variants of concern. However, the expansion of a novel variant descended from cluster 20C and designated CAL.20C (20C/S:452R or B.1.429) has been reported in Southern California [45]. CAL.20C variant was first observed in July 2020 in one of 1247 samples from Los Angeles County and was not detected in Southern California again until October. Since then, the prevalence of this variant has increased, and, in January 2021, it accounted for 35% and 44% (37 of 85) of all samples collected in California state and Southern California, respectively. However, relatively few samples have been sequenced, and sequencing is not performed uniformly throughout the state, making it difficult to establish a more accurate estimate of the expansion of this variant [46]. CAL.20C variant is defined by five mutations (ORF1a: I4205V; ORF1b: D1183Y; S: S13I, W152C, and L452R). In particular, the L452R mutation in the spike protein has been found to be resistant to certain therapeutic monoclonal antibodies [47]. As clinical outcomes have yet to be established, the functional effect of CAL.20C variant regarding infectivity and disease severity remains uncertain.

Another new coronavirus variant, named A.23.1, has been detected in Uganda and has quickly become the most common coronavirus in Uganda’s capital city, Kampala. The set of the spike mutations in A.23.1 includes R102I, F157L, V367F, Q613H, and P681R. Additional substitutions in non-spike regions include non-structural protein (nsp) 3: E95K; nsp6: M86I and L98F; ORF8: L84S and E92K, and N: S202N and Q418H. As of 16 February 2021, 274 sequences of A.23.1 lineage have been detected in 17 countries [48]. In addition, an emerging lineage (now designated as B.1.526) of viral isolates in the New York region that shares mutations with previously reported variants has been recently detected by West et al. using a tool to query the spike mutational landscape. The most common sets of spike mutations in B.1.526 are L5F, T95I, D253G, and E484K or S477N, D614G, and A701V. This lineage appeared in late November 2020, and it accounts for ~5% of coronavirus genomes sequenced and was deposited in Global Initiative on Sharing Avian Influenza Data (GISAID) during late January 2021 [49]. Although the clinical impact of the A.23.1 and the B.1.526 variants is not yet clear, it is essential to perform a careful monitoring of these variants as well as a rapid assessment of the consequences of the spike protein changes for vaccine efficacy.

The UK has strengthened genomic surveillance to evaluate the molecular evolution of the prevalent B.1.1.7 variant. New variants with different substitutions have emerged as a consequence of both the high replication rates of the virus and the increasing selection pressure resulting from the growth of the seroprevalent fraction of the population of England. The ones that worry the most are L18F and E484K. The introduction of the L18F mutation confers a replicative advantage to the virus [50], whereas E484K mutation could confer resistance to immunity. Moreover, other non-B.1.1.7 lineages with the E484K mutation have been identified in some UK regions such as the VUI 202102/01 (A.23.1 with E484K) or the B.1.525 (VUI 2021 02/03) with 4 mutations within the spike protein (Q52R, E484K, Q677H, and F888L). Further work is needed to establish the impact of these mutations on protective vaccines efficacy in the context of the evolving variants that have acquired E484K mutation [51].

There is substantial variability in the course of COVID-19, ranging from asymptomatic infection to death. One of the main topics of ongoing research is how the emergence of the new SARS-CoV-2 variants impacts patient’s outcome. However, there are no consistent data published yet, in part due to the fact that most of the genome sequences shared are not linked to clinical outcomes. One study performed by researchers from the University of Washington comparing two dominant clades of virus in circulation showed no significant difference in outcomes of hospitalization or death between clades [52]. Similarly, clinicians and scientists working in the frontline in South Africa have not observed any differences in symptoms in people infected with the new variant P.1.351, compared with people infected with other variants [53]. Nevertheless, further analysis is necessary to screen differences in COVID-19 symptom type, severity, or duration of the disease caused by the new VOC.

## 4. Population Monitoring of Variant by Genomic Sequencing

Worldwide expansion of genomic sequencing and data exchange is essential to detect the emergence of new variants or their introduction in a given country or region. To date, more than 528,000 sequences have been submitted to the GISAID that promotes the rapid sharing of data from the coronavirus causing COVID-19; however, most of them come from only a few countries. It is necessary that all the countries share sequence information to understand the spread of SARS-CoV-2. Improving the geographical coverage of sequencing is essential for the world to adequately capture the viral changes and establish alternative measures, as previously reported in Netherlands [54]. Increased sequencing capacity is a priority research area for the WHO. In order to achieve this objective, a number of recommendations have been established highlighting the need to have or implement a network of laboratories with sequencing experience integrated within the epidemiological surveillance system to generate useful information for the decision-making of public health measures and to ensure that resources are available to manage increasing numbers of COVID-19 detection and characterization of sample requests.

## 5. Defining SARS-CoV-2 Variants More Resistant to Vaccine Action

There are several lines of action to establish the role played by the mutations introduced into the SARS-CoV-2 genome on resistance to the action of the immune responses induced by current vaccines mostly targeting the spike protein. To this end, the scientific community, WHO, and companies have warned of the need to experimentally establish assays that determine to what extent the various variants already identified and those upcoming, that are spreading in the population, are due to mutational changes that confer an enhanced degree of resistance to the antibodies generated in infected or vaccinated people. Experimental approaches to be followed are: (1) demonstration in experimental animal models (humanized mice susceptible to virus, hamsters, ferrets, and macaques) that emerging variants are more virulent (higher transmission, replication, organ damage, morbidity, and mortality) than Wuhan’s reference strain; (2) demonstration that these variants withstand the neutralizing action of immune sera induced in current vaccination campaigns, such as mRNA-based vaccines (Pfizer, Moderna), non-replicating adenovirus-based vaccines (AstraZeneca, Janssen, Sputnik), protein subunit (Novavax), inactivated virus (Sinovac), and others; and (3) demonstration of the degree of variant control in vaccinated personnel in comparison with infection by the parental Wuhan strain

In those cases in which increased resistance to antibodies was demonstrated, as indicated by the above experimental data, it should also be confirmed that T cell responses are also affected, or if, on the contrary, the controlling effect of the infection by T lymphocytes is maintained. These experimental data in animals and humans are necessary to establish the highest sensitivity and resistance of the different variants to the immune action.

The reduction in the efficacy of different vaccine candidates in regions where the variants of concern have become prevalent has accelerated the decision of many of the companies that produce current COVID-19 vaccines to consider modifying their designs to cover circulating viral variants. The immediate consequence is that there should be on the market new vaccines from the same production companies, which could be given either to those people already vaccinated as a recall dose or to those who have not yet received the vaccine dose. These new vaccines modified in their specific platforms would entail additional costs and could in turn lead to more resistant variants with additional mutations due to selective pressure from the immune system. We hope this does not occur, but we must remain vigilant about the evolution.

Anything that can be done to suppress spread of SARS-CoV-2 will help to limit the emergence of new variants. However, other strategies that include multivalent designs focused on conserved regions of different viral proteins could be of great prophylactic relevance to counteract escape variants that are emerging. In this regard, a sustained effort to develop a pan-SARS-CoV-2 vaccine is warranted. Similarly, to improve the efficacy of the current treatments, it will be of relevance to use a combination of antibodies directed against other viral regions in addition to the spike protein or to implement the “lethal mutagenesis” strategy as alternatives that slow the viral diversification [55].

## 6. Concluding Remarks

The appearance of variants of SARS-CoV-2 is not an unexpected virological finding but rather a result of natural selection, giving rise to mutations within the long-RNA (30,000 nucleotides) sequence of coronaviruses. In spite of a viral proof-reading exonuclease, multiple variants emerge in the population, but only those with an advantage on virus replication and dissemination prevail. This is further enhanced by the large incidence of infection rates within humans, accentuated by the presence within the viral genome of genes encoding antagonists of host defense mechanisms, such as blocking interferon action and other immune stimulatory molecules. The virus tries to counteract the host response, giving rise to mutations. Within a year since the virus first appeared in China and its rapid spread, we are confronted with the emergence of variants of concern in different parts of the world. The main characteristics are higher binding affinity for the cellular ACE-2 receptor than the parental Wuhan virus, the enhanced resistance to neutralizing antibodies, and increased virulence.

To promote actions for the control of the emerging variants, a major effort is being put forward by different nations and institutions, such as WHO, CEPI (Coalition for Epidemic Preparedness Innovations), Gates Foundation, GAVI (Global Alliance for Vaccines and Immunizations), and others, with the purpose to make universal access to vaccines and to assure control of virus infection. Indeed, we have proven that current and incoming vaccines will cope with the control of variants and the potential eradication of the virus. In this regard, the results coming out from Israel on the high efficacy of the Pfizer vaccine against SARS-CoV-2 infections are encouraging, in a country where the UK variant is prevalent. It will be only through the detailed understanding of the virus structure, biology, and vaccine developments that we can finally achieve the control of SARS-CoV-2 infections.

## Figures and Tables

**Table 1 vaccines-09-00243-t001:** Main characteristics of the emergent severe acute respiratory syndrome coronavirus-2 (SARS-CoV-2) variants. Data updated on February 16 2021 (https://outbreak.info (accessed on 9 March 2021)).

Variants	B.1.1.7	B.1.351	P.1
1st detection	September 2020	8 October 2020	2 January 2021
Detection site	United Kingdom	South Africa	Japan/Brazil
Mutations in S protein	7 mutations: **N501Y**, A570D, **D614G**, P681H, T716I, S982A, D1118H 2 deletions: H69-V70del, Y144del	9 mutations: L18F, D80A, D215G, R246I, K417N, E484K, **N501Y**, **D614G**, A701V 1 deletion: LAL 242-244 del	12 mutations: L18F, T20N, P26S, D138Y, R190S, K417T, E484K, **N501Y**, **D614G**, H655Y, T1027I, V1176F
Countries reported cases	82	40	19
Countries with sequences	64	35	14
Potential risk	- Higher transmission - Higher disease severity - Modest reduction in the neutralization efficacy of sera from convalescent patients or vaccinees	- Higher transmission - Higher reinfection rates - Significant reduction in the neutralization efficacy of sera from convalescent patients or vaccinees	- Higher transmission - Higher reinfection rates - Significant reduction in the neutralization efficacy of sera from convalescent patients or vaccinees

## Data Availability

Not applicable.

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
