# Peer review of "Emerging SARS-CoV-2 Variants and Impact in Global Vaccination Programs against SARS-CoV-2/COVID-19"

_vaccines, 2021, doi:10.3390/vaccines9030243_

Round 1

Reviewer 1 Report

The article is a review about the newly emerging variants of
SARS-CoV-2 based on the very recent data, especially the highly
transmissible three variants: B.1.1.7 (appeared in UK, Dec. 2020),
B.1.351 (appeared in South Africa, Dec. 2020, and P.1
(appeared in Japan, Jan. 2021). These variants demonstrate higher
pathogenic and spread exceptional fast, thus attract global
attention. The article focused on the mutations in the s
pike (S) gene and raised potential risks to COVID-19 prevention,
for example, the mutations in these variants may cause a
reduction of the efficacy of vaccination used presently,
and threaten the final control of COVID-19. The article
is well organized and supported with sufficient evidence.
It can be published in the present format.

Author Response

Reviewer 1

The article is a review about the newly emerging variants of SARS-CoV-2 based on the very recent data, especially the highly transmissible three variants: B.1.1.7 (appeared in UK, Dec. 2020), B.1.351 (appeared in South Africa, Dec. 2020) and P.1 (appeared in Japan, Jan. 2021). These variants demonstrate higher pathogenic and spread exceptional fast, thus attract global
attention. The article focused on the mutations in the spike (S) gene and raised potential risks to COVID-19 prevention; for example, the mutations in these variants may cause a reduction of the efficacy of vaccination used presently and threaten the final control of COVID-19. The article is well-organized and supported with sufficient evidence. It can be published in the present format.

Answer: We thank the reviewer for his/her favourable comments.

Reviewer 2 Report

I was invited to revise the paper entitled "Emerging SARS-CoV-2 variants and impact in global vaccination programs against SARS-CoV-2/COVID-19". It is a review that aimed to syntetize the actual evidences about most circulating Sars-cov-2 variants impact on vaccine efficacy.

I want to congratulate with Authors for the excellent work and fot the deep analysis. It is an outstanding work that can improve the knowledge of this topic.

The introduction is well presented. Single variant sections deeply describe variants characteristics, linking epidemiological and genetic notions. Table 1 is informative and summarize all main aspects of the text.

In my opinion Authors should focus the discussion also on the impact of variants on patients outcomes.

Minor observation: References were reported with a different font from text.

Author Response

Reviewer 2 

I was invited to revise the paper entitled "Emerging SARS-CoV-2 variants and impact in global vaccination programs against SARS-CoV-2/COVID-19". It is a review that aimed to synthesize the actual evidences about most circulating SARS-CoV-2 variants impact on vaccine efficacy.

I want to congratulate with Authors for the excellent work and for the deep analysis. It is an outstanding work that can improve the knowledge of this topic.

The introduction is well presented. Single variant sections deeply describe variants characteristics, linking epidemiological and genetic notions. Table 1 is informative and summarize all main aspects of the text.

Answer: We thank the reviewer for his/her favourable comments.

In my opinion Authors should focus the discussion also on the impact of variants on patients’ outcomes.

Answer: Although the literature already published on this topic does not allow to perform a critical discussion on the impact of variants on patient’s outcome, we have introduced the following paragraph and references in the text (lines 433-445):

There is substantial variability in the course of COVID-19, ranging from asymptomatic infection to death. One of the main topics of ongoing research is how the emergence of the new SARS-CoV-2 variants impacts on patient’s outcome. However, there are no consistent data published yet, in part due to the fact that most of the genome sequences shared are not linked to clinical outcomes. One study performed by researchers from the University of Washington comparing two dominant clades of virus in circulation showed no significant difference in outcomes of hospitalization or death between clades [1]. Similarly, clinicians and scientists working in the frontline in South Africa have not observed any differences in symptoms in people infected with the new variant P.1.351, compared with people infected with other variants [2]. Nevertheless, further analysis is necessary to screen differences in COVID-19 symptom type, severity, or duration of the disease caused by the new VOC.

  1. Nakamichi, K.; Shen, J.Z.; Lee, C.S.; Lee, A.; Roberts, E.A.; Simonson, P.D.; Roychoudhury, P.; Andriesen, J.; Randhawa, A.K.; Mathias, P.C.; et al. Hospitalization and Mortality Associated with SARS-CoV-2 Viral Clades in COVID-19. Sci. Rep. 2021, 11, doi:10.1038/s41598-021-82850-9.
  2. Oliveira, T. de; Hanekom, W. South African Scientists Who Discovered New COVID-19 Variant Share What They Know Available online: http://theconversation.com/south-african-scientists-who-discovered-new-covid-19-variant-share-what-they-know-153313 (accessed on 4 March 2021).

Minor observation: References were reported with a different font from text.

Answer: The references were modified as suggested.

Reviewer 3 Report

This review is a well organized discussion of the current issue of variants in the SARS-COV-2 pandemic. The topic brought multiple sources on the current variants and an addition of the next vaccines coming. However extensive English editing is required.

My comments are the following:

  • Their modes of action of different types of currently available COVID-19 vaccines such as traditional, mRNA, and adenovirus ones need to be explained in detail as separate paragraphs before description of the variants and protection against these.
  • I recommend including a list of all abbreviations used in the text and paying attention to write the full names of the acronyms reported in the text.
  • The authors should cite the following review (Palma G, Pasqua T, Silvestri G, et al. PI3Kδ Inhibition as a Potential Therapeutic Target in COVID-19. Front Immunol. 2020;11:2094. Published 2020 Aug 21. doi:10.3389/fimmu.2020.0209) and discuss how these variants can affect the Italian pandemic situation?

Author Response

Reviewer 3

This review is a well-organized discussion of the current issue of variants in the SARS-CoV-2 pandemic. The topic brought multiple sources on the current variants and an addition of the next vaccines coming. However, extensive English editing is required.

Answer: We have carefully checked the English in the manuscript.

My comments are the following:

Their modes of action of different types of currently available COVID-19 vaccines such as traditional, mRNA, and adenovirus ones need to be explained in detail as separate paragraphs before description of the variants and protection against these.

Answer: The following paragraph and references were included in the introduction of the manuscript (lines 51-95):

The COVID-19 pandemic has required rapid action and the development of vaccines in an unprecedented timeframe. According to WHO (https://www.who.int/publications/m/item/draft-landscape-of-covid-19-candidate-vaccines), 76 vaccine candidates based on several different platforms are being currently evaluated in human clinical trials while 182 candidates are under investigation in preclinical models. The four SARS-CoV-2 vaccines licensed at present by the regulatory agencies are based on nucleic acid or non-replicating viral vectored platforms.

The two vaccines based on messenger ribonucleic acid (mRNA) have been developed by Moderna (mRNA-1273) and Pfizer/BioNTech (BNT162b2) pharmaceutical companies and contain the genetic information for the synthesis of the stabilized pre-fusion form of the SARS-CoV-2 Spike (S) protein encapsulated in a lipid nanoparticle (LNP) vector that enhances uptake by host immune cells. These vaccines use the host cell transcription and translation machinery to produce the viral S protein that is processed afterward and recognized by specific B and T cells eliciting both humoral and cellular adaptive immune responses able to confer protection against COVID-19 illness, including severe disease. The reported efficacy of a two-dose regimen of the mRNA-1273 or BNT162b2 vaccines is 94.1% [3] or 95% [4], respectively.

The two other licensed vaccines have been developed by Oxford University/AstraZeneca (AZD1222) and Janssen (Ad26.COV2.S) pharmaceuticals and are based on two different modified non-replicating adenoviruses. The AstraZeneca candidate is a monovalent vaccine composed of a single recombinant, replication-deficient chimpanzee adenovirus (ChAdOx1) vector encoding the S glycoprotein of SARS-CoV-2. The S protein is expressed in the trimeric pre-fusion conformation. Following administration, the S glycoprotein of SARS-CoV-2 is expressed locally stimulating neutralizing antibody and cellular immune responses, which may contribute to protection to COVID-19. The AZD1222 vaccine has an efficacy of 63.09% against symptomatic SARS-CoV-2 infection. Vaccine efficacy was 62.6% in participants receiving two recommended doses with any dose interval (ranging from 3 to 23 weeks) [5]. The Janssen vaccine is based on the adenovirus serotype 26 (Ad26) which expresses the stabilized pre-fusion SARS-CoV-2 S protein. As opposed to the ubiquitous Ad5 serotype, very few people have been exposed to the rare Ad26 serotype; therefore, pre-existing immunity against the vector reducing this candidate’s immunogenicity is not expected to be a major concern. A Phase 3 randomized and placebo-controlled trial of the single-dose Ad26.COV2.S in approximately 40,000 participants is currently ongoing. The primary analysis of 39,321 participants using a data cut-off date of January 22, 2021 demonstrated a vaccine efficacy of 66.9%.

The fifth vaccine waiting for approval has been developed by Novavax Company (NVX-CoV2373). It is a protein subunit vaccine constructed from the full-length, stabilized pre-fusion SARS-CoV-2 S glycoprotein, produced in the established Sf9 insect cell expression system and adjuvanted by saponin-based Matrix M1 [6]. In January, Novavax announced that in the British trial, the vaccine had an efficacy rate of 89%. Since all the vaccines that have been administered worldwide are focused on the spike protein, which accumulates high rate of mutations during viral evolution, as evidenced in the genome sequences from the new emerging SARS-CoV-2 variants, it is imperative to evaluate the impact of those mutations on the actual efficacy of COVID-19 vaccines.

  1. Baden, L.R.; El Sahly, H.M.; Essink, B.; Kotloff, K.; Frey, S.; Novak, R.; Diemert, D.; Spector, S.A.; Rouphael, N.; Creech, C.B.; et al. Efficacy and Safety of the MRNA-1273 SARS-CoV-2 Vaccine. N. Engl. J. Med. 2021, 384, 403–416, doi:10.1056/NEJMoa2035389.
  2. Polack, F.P.; Thomas, S.J.; Kitchin, N.; Absalon, J.; Gurtman, A.; Lockhart, S.; Perez, J.L.; Pérez Marc, G.; Moreira, E.D.; Zerbini, C.; et al. Safety and Efficacy of the BNT162b2 MRNA Covid-19 Vaccine. N. Engl. J. Med. 2020, 383, 2603–2615, doi:10.1056/NEJMoa2034577.
  3. Voysey, M.; Clemens, S.A.C.; Madhi, S.A.; Weckx, L.Y.; Folegatti, P.M.; Aley, P.K.; Angus, B.; Baillie, V.L.; Barnabas, S.L.; Bhorat, Q.E.; et al. Safety and Efficacy of the ChAdOx1 NCoV-19 Vaccine (AZD1222) against SARS-CoV-2: An Interim Analysis of Four Randomised Controlled Trials in Brazil, South Africa, and the UK. The Lancet 2021, 397, 99–111, doi:10.1016/S0140-6736(20)32661-1.
  4. Tian, J.-H.; Patel, N.; Haupt, R.; Zhou, H.; Weston, S.; Hammond, H.; Logue, J.; Portnoff, A.D.; Norton, J.; Guebre-Xabier, M.; et al. SARS-CoV-2 Spike Glycoprotein Vaccine Candidate NVX-CoV2373 Immunogenicity in Baboons and Protection in Mice. Nat. Commun. 2021, 12, 372, doi:10.1038/s41467-020-20653-8.
  • I recommend including a list of all abbreviations used in the text and paying attention to write the full names of the acronyms reported in the text.

Answer: Following the reviewer recommendations, we have written the full names of the acronyms the first time they appear in the text.

  • The authors should cite the following review (Palma G, Pasqua T, Silvestri G, et al. PI3Kδ Inhibition as a Potential Therapeutic Target in COVID-19. Front Immunol. 2020;11:2094. Published 2020 Aug 21. doi:10.3389/fimmu.2020.0209) and discuss how these variants can affect the Italian pandemic situation.

Answer: The suggested review is out of the main objective of our manuscript, which is to describe the different SARS-CoV-2 variants of concern that have been identified as well as their impact in the global vaccination programs against SARS-CoV-2/COVID-19.